# CRISPR targeting of H3K4me3 activates gene expression and unlocks centromere-proximal crossover recombination in Arabidopsis

Jenia Binenbaum [1,6], Vanda Adamkova[1,6], Hannah Fryer [1], Linhao Xu [1], Nicola Gorringe [1], Piotr Włodzimierz[1,4], Robin Burns[1], Ashot Papikian [2,5], Steven E. Jacobsen [2,3], Ian R. Henderson [1] & C. Jake Harris [1] ✉

H3K4me3 is a fundamental and highly conserved chromatin mark across eukaryotes, playing a central role in many genome-related processes, including transcription, maintenance of cell identity, DNA damage repair, and meiotic recombination. However, identifying the causal function of H3K4me3 in these diverse pathways remains a challenge, and we lack the tools to manipulate it for agricultural benefit. Here we use the CRISPR-based SunTag system to direct H3K4me3 methyltransferases in the model plant, *Arabidopsis thaliana*. Targeting of SunTag-SDG2 activates the expression of the endogenous reporter gene, *FWA*. We show that SunTag-SDG2 can be employed to increase pathogen resistance by targeting the H3K4me3-dependent disease resistance gene, *SNC1*. Meiotic crossover recombination rates impose a limit on the speed with which new traits can be transferred to elite crop varieties. We demonstrate that targeting of SunTag-SDG2 to low recombining centromeric regions can significantly stimulate proximal crossover formation. Finally, we reveal that the effect is not specific to SDG2 and is likely dependent on the H3K4me3 mark itself, as the orthogonal mammalian-derived H3K4me3 methyltransferase, PRDM9, produces a similar effect on gene expression with reduced off-target potential. Overall, our study supports an instructive role for H3K4me3 in transcription and meiotic recombination and opens the door to precise modulation of important agricultural traits.

Re-programming of transcriptional landscapes in response to developmental and environmental cues critically underlies a plant's ability to adapt and optimise its performance. The potential to rewire and fine-tune the expression of endogenous genes, therefore, holds great promise for agricultural improvement. Chromatin modifications associated with gene expression play an essential role in this process[1].

Trimethylation of histone 3 lysine 4 (H3K4me3) is a well-studied and highly conserved chromatin mark. H3K4me3 enrichment around the transcriptional start site of expressed genes is observed across

[1]Department of Plant Sciences, University of Cambridge, Cambridge, UK. [2]Department of Molecular, Cell and Developmental Biology, University of California, Los Angeles, CA, USA. [3]Howard Hughes Medical Institute, University of California, Los Angeles, CA, USA. [4]Present address: Institute of Biochemistry and Biophysics, Polish Academy of Sciences, Warsaw, Poland. [5]Present address: Plant Molecular and Cellular Biology Laboratory, Salk Institute for Biological Studies, La Jolla, CA, USA. [6]These authors contributed equally: Jenia Binenbaum, Vanda Adamkova. ✉e-mail: cjh92@cam.ac.uk

eukaryotes[2], and is thought to assist in transcriptional maintenance and memory of cell identity[3]. Despite the ubiquity of H3K4me3 and its positive correlation with gene expression, whether this chromatin mark plays an instructive role in transcription remains a matter of debate, due in part to the limited evidence of transcriptional impact when H3K4me3 levels are perturbed[4,5]. In animals, TAF3 directly binds to H3K4me3 and recruits basal transcription factor machinery, providing a plausible mechanism to stimulate transcription[6,7]. Recent work in embryonic stem cells shows that H3K4me3 plays a role in RNA Polymerase II pause-release and elongation, rather than transcriptional initiation[8]. In plants, there are no TAF3 homologues, and whether similar mechanisms exist to promote transcription downstream of H3K4me3 remains unknown[9–13].

H3K4me3 also plays a key role in several other genome-related processes. For instance, meiotic recombination, which allows for genetic exchange between parental chromosomes[14]. This process is essential for breeders to transfer traits from wild relatives into elite cultivars[15,16]. However, the positioning of meiotic recombination events is non-uniform across the genome, with gene-rich chromosomal arms having relatively higher rates of crossover recombination, and centromeric repeat-rich regions being more recombination-suppressed[17,18]. Loci encoding desirable traits that reside within these low-recombining regions of the genome can show linkage drag with deleterious variants during traditional breeding approaches, which limits genetic gains[15,16]. In mammals, the H3K4me3 SET domain methyltransferase containing protein, PRDM9, helps to define active sites of crossover recombination[19]. While H3K4me3 levels are positively correlated with recombination rates in plants[17,20], whether deposition of H3K4me3 is sufficient to promote crossover formation is entirely unknown.

One way to reveal the causal role of an epigenetic modification is to directly deposit the mark at targeted genomic locations[21–25]. Epigenome editing tools, such as the CRISPR-based SunTag system[26], have proven to be highly efficient at targeting DNA methylation addition, removal, and histone modification[23,27–32], although their utility for modulating agronomically useful traits is only just beginning to be realised[33], and is limited by the range of possible marks that can be directed[1]. In the model plant *Arabidopsis thaliana*, SDG2 is the main histone methyltransferase responsible for deposition of H3K4me3[34]. Here, we describe CRISPR-directed H3K4me3 deposition in plants and demonstrate its utility in a range of scenarios. We translationally fuse the SDG2 methyltransferase domain to the scFv single-chain antibody, which can be recruited to the SunTag epitope chain connected to a dead Cas9 (dCas9). We show that SunTag-SDG2-mediated H3K4me3 deposition can activate the transcription of endogenous genes. Further, by targeting a disease resistance gene, we generate plants with enhanced resistance capabilities. Next, we show that SunTag-SDG2 can unlock meiotic crossover recombination in centromere-proximal regions. Finally, we incorporate the methyltransferase domain from PRDM9 into SunTag - which was recently shown to instruct transcription in a mammalian context[31] - showing that it is also highly efficient for transcriptional activation in plants, and displays reduced off-target potential. These results demonstrate the power and versatility of a functional H3K4me3 targeting system for modulation of critical genome templated processes, such as transcriptional enhancement and meiotic recombination.

## Results
### SunTag-SDG2 mediates transcriptional activation of *FWA*
The SunTag system we chose is composed of three components[26,35] (Fig. 1a): (1) A catalytically deactivated (nuclease-deficient) Cas9 (dCas9) translationally fused to a tail with 10× copies of the GCN4 epitope, each separated by a 22 amino acid flexible linker. (2) An effector module that encodes a single-chain fragment variable (scFv) antibody that recognises GCN4, superfolder GFP (sfGFP), and the

effector of interest (e.g., enzymatic modifier/recruitment scaffold). (3) *U6* promoter-driven CRISPR guide RNA. When the three components are expressed within the same cell, the guide RNA recruits the dCas9-10xGCN4 to the target locus of interest, while the GCN4 single-chain antibody containing the effector module binds to the dCas9-10xGCN4 epitope tail. The system allows for conformational flexibility and the potential for high stoichiometric concentration of the effector at the locus of interest.

We inserted the C-terminal coding sequence of the H3K4 methyltransferase, SDG2 (amino acids 1571–2335), into the SunTag system (Fig. 1b, Supplementary Fig. S1). As a control, we also generated a version of SDG2 with an amino acid change (Y1903F) predicted to abolish catalytic activity (Supplementary Fig. S2)[34,36]. Hereafter, we refer to these as SDG2 and dSDG2 (for catalytically deactivated). We targeted the SunTag system to the promoter of the epigenetically repressed *FWA* gene[37], using the previously published CRISPR guide RNA 4, which is complementary to two tandem repeat regions directly upstream of the *FWA* transcription start site (TSS)[29]. These were transformed into *rdr6*, as previous results have shown that SunTag is less prone to silencing and achieves higher transgene expression in this background[27]. We observed that SunTag:SDG2:FWA_g4 activated *FWA* mRNA expression, while the SunTag:dSDG2:FWA_g4 and no-guide RNA (SunTag:SDG2:No_g) controls did not (Fig. 1c, Supplementary Fig. 1C). Importantly, effector module expression was detected in all constructs, indicating that both guide RNA targeting and catalytic activity are required for *FWA* activation (Supplementary Fig. 3A). To validate the presence of the SunTag and test for the deposition of H3K4me3, we performed ChIP qPCR and observed significant enrichment both for the presence of SunTag and for H3K4me3 accumulation at the target site in these transgenic lines (Fig. 1d and Supplementary Fig. 3B), confirming that SunTag-SDG2 binds the *FWA* locus and deposits H3K4me3. To investigate SunTag:SDG2:FWA_g4 binding genome-wide, we performed high-throughput sequencing (anti-HA ChIP-seq; the HA-tag is present on both the dCas9 and effector fusion modules), which confirmed the previously reported high specificity of SunTag targeting[28], with the *FWA* target locus as the most enriched binding target (Fig. 1e, Supplementary Fig. 4). Only 3 conserved binding sites were identified in the SunTag:SDG2:FWA_g4 lines; *FWA* and two off-target sites previously identified as having sequence-similarity to FWA_g4[28]. Furthermore, transcriptome analyses confirmed *FWA* as the highest fold change differentially upregulated gene (Fig. 1f). Together, these data indicate that deposition of H3K4me3 by SunTag-SDG2 is sufficient to overcome epigenetic silencing and drive transcriptional activation in a locus-specific manner.

Next, to examine the genome-wide effect on H3K4me3 by SunTag-SDG2, we performed ChIP-seq analysis, confirming enrichment at the *FWA* locus and canonical genic H3K4me3 patterning (Fig. 1e, Supplementary Fig. 5). In addition to a modest peak at the +1 nucleosome at *FWA*, the H3K4me3 signal extended into the gene body. Performing metaplots over genes, transcriptional start sites (TSS's), and endogenous H3K4me3 peaks, we observed that SunTag-SDG2 lines generally display a moderate increase of H3K4me3 over the 3' end of gene bodies (Fig. 1g). Correspondingly, we observed reduced H3K4me3 signal at endogenous peaks and TSS regions (Fig. 1g). As the H3K4me3 peak at *FWA* in our SunTag:SDG2:FWA_g4 lines was not as strong as that observed for *fwa* in the epiallelic state[27,38], we asked whether the effect could persist in the next generation. Analysis of $T_2$ seedlings from two independent lines segregating for the presence of SunTag:SDG2:FWA_g4 showed that *FWA* activation was entirely dependent on the presence of the transgene (Supplementary Fig. 6). Together these data indicate that while SunTag:SDG2:FWA_g4 can site-specifically activate *FWA* expression, it is insufficient to overcome the endogenous silencing machinery as required to switch *FWA* into the stably active *fwa* epiallelic state[32,39].

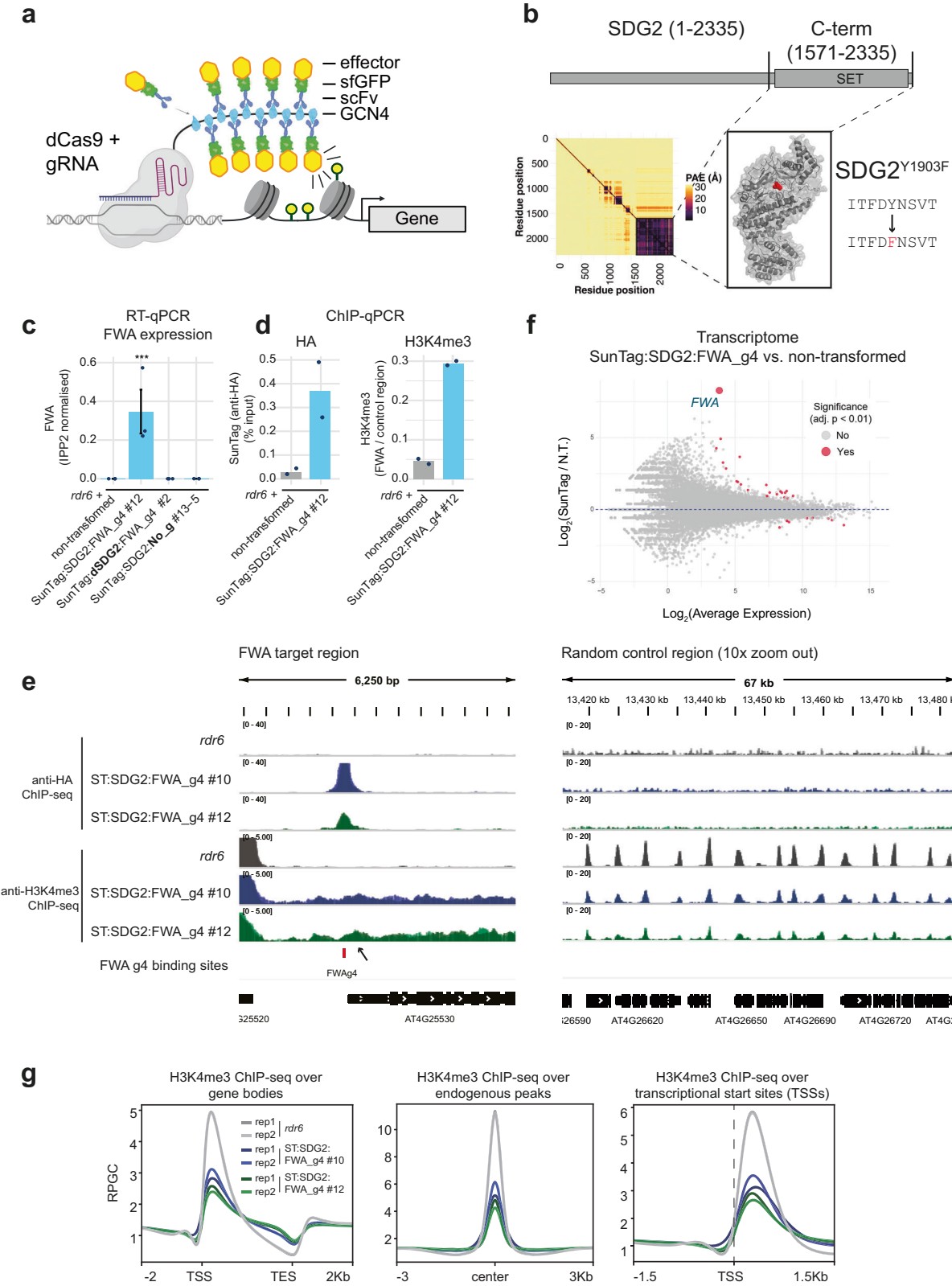

**a**

**b** SDG2 (1-2335) · C-term (1571-2335)

**c** RT-qPCR FWA expression

**d** ChIP-qPCR

**f** Transcriptome SunTag:SDG2:FWA_g4 vs. non-transformed

**e** FWA target region · Random control region (10x zoom out)

**g** H3K4me3 ChIP-seq over gene bodies · H3K4me3 ChIP-seq over endogenous peaks · H3K4me3 ChIP-seq over transcriptional start sites (TSSs)

## SNC1-mediated modulation of disease resistance

Given that SunTag-SDG2 can modulate chromatin and activate *FWA* gene expression, we were interested in using the tool to modify an important plant trait. Gene expression during pathogen attack is a key factor in determining disease outcome. *SNC1* is a disease resistance gene, encoding a nucleotide-binding leucine-rich repeat protein[40] that is located within the partially epigenetically repressed *RPP5* gene cluster[41–43]. *SNC1* expression activity is modulated by H3K4me3[44], and its level of expression is positively correlated with basal resistance[41,44]. Therefore, *SNC1* is an attractive target for proof-of-principle modulation of disease resistance by epigenome engineering.

We designed guide RNAs to target the promoter region of *SNC1* directly upstream of the TSS and inserted them into our SunTag-SDG2 and SunTag-dSDG2 constructs (Fig. 2a, Supplementary Fig. 7). As high

**Fig. 1 | SunTag-SDG2 activates *FWA* mRNA expression. a** Schematic showing the SunTag system for modulation of chromatin[28,35]. Created in BioRender. **b** The upper panel shows SDG2 with the catalytic SET domain (drawn approximately to scale). Lower left depicts predicted alignment error (PAE) plot from AlphaFold3[75] prediction of the SDG2 coding sequence with a box drawn around the region cloned into SunTag (amino acids 1571–2335). Lower right depicts the AlphaFold3 model for SDG2$_{1571-2335}$, with the position of the amino acid change (Y1903F) for dSDG2 indicated (shown in red). **c** RT-qPCR for *FWA* expression from the genotypes indicated. Error bars represent SEM from three biological replicates (a pool of 10 seedlings) from the lines (#) and genotypes indicated. *** indicates $p < 0.005$ (ANOVA with post-hoc Tukey HSD. **d** ChIP qPCR for the presence of SunTag (left panel) and H3K4me3 enrichment (right panel) over the TSS of *FWA*. The control region (AT5G65130) used for normalisation has high H3K4me3. **e** Genome browser images showing ChIP-seq enrichment of SunTag (anti-HA) and H3K4me3 levels of two independent SunTag:SDG2:FWA_g4 lines (2 biological replicates each, bio replicate tracks are overlaid) at the *FWA* TSS (left panel) and at a random genic control region. **f** MA plot comparing the transcriptome of SunTag:SDG2:FWA_g4 as compared to the non-transformed control (*rdr6*). Differentially expressed genes were defined using an adjusted *p*-value (Bonferroni method), shown in red (FDR < 0.01), with *FWA* labelled and enlarged for visibility. **g** H3K4me3 ChIP-seq metaplots over all genes, H3K4me3 endogenous peaks, and TSS regions.

pathogen resistance is generally associated with dwarfed plants, we inspected the lines for growth and developmental phenotypes. In our *SNC1* targeting SunTag-SDG2, but not our SunTag:dSDG2 control, we observed a small, dwarfed phenotype with reduced rosette surface area (Fig. 2a, b, Supplementary Fig. 7). These plant phenotypes resembled those of our positive control *bal* lines, which overexpress *SNC1*, resulting in the small, dwarfed plant phenotype and high basal disease resistance[41]. As the phenotype is consistent with upregulation of *SNC1* by SunTag, we confirmed the increased expression and successful targeting of SunTag-SDG2 to the *SNC1* locus (Fig. 2c, d, Supplementary Fig. 7).

To assess whether these lines show increased pathogen resistance, we challenged the plants with the generalist plant pathogen, *Pseudomonas syringae* pathovar *tomato* strain DC3000 (*Pst*). The recently described bioluminescent *Pst* (*Pst::LUX*) was used to allow for non-destructive quantification of pathogen colonisation[45]. We developed a high-throughput assay system, inoculating 7-day-old seedlings with *Pst::LUX* and imaging to quantify the levels of *Pst::LUX* at 24 h intervals over the course of infection. We challenged a range of control genotypes that are known to be *Pst* hyper-resistant (*bal*), constitutively primed (*edr1*), and susceptible (*NahG*), recapitulating the expected pattern of *Pst::LUX* colonisation dynamics in these backgrounds (Supplementary Fig. S8)[45]. Next, we challenged our *SNC1* targeting lines with *Pst::LUX*, finding that they displayed hyper-resistance that was nearly identical in magnitude to that of the *SNC1* overexpressing positive control, *bal* (Fig. 2e, Supplementary Fig. 7). The results demonstrate that epigenome engineering of a single defence gene, *SNC1*, is sufficient to generate plants with improved disease resistance phenotypes in Arabidopsis.

**Centromeric targeting of SDG2 drives increased meiotic crossover recombination**

Beyond transcription, H3K4me3 is associated with many other fundamental genome-related processes, including meiotic recombination. In plants, meiotic crossover recombination rates broadly correlate with epigenetic territories and are non-uniform across the genome[17,18]. In *A. thaliana*, recombination occurs at a relatively high frequency in euchromatic arms, whereas it is significantly reduced over the heterochromatic pericentromeric and centromeric[46–48]. Crossover hotspots are often found to be high at gene promoters where H3K4me3 is enriched[20]. In previous work, we showed that gene-associated crossover recombination hotspots can be suppressed by small RNA-directed DNA methylation[49]. Therefore, we hypothesised that recombination within the centromeric regions might be increased by SunTag-mediated epigenetic activation via deposition of H3K4me3.

The Arabidopsis centromeres are composed of highly repetitive *CEN178* satellite repeat arrays[47,50]. We designed a guide RNA that perfectly matches 241 target sites within the *CEN178* repeats within the centromeric region of chromosome 3 (LRCen3_g), and inserted this guide into the SunTag-SDG2 system. As centromere-proximal crossover recombination events are relatively rare, to quantitatively assess the impact on recombination rate over centromere 3, we crossed (non-transgenic) Col-0 plants with the *CTL3.9* crossover reporter[51]. *CTL3.9* is

a transgenic line that encodes linked T-DNAs that encode red and green fluorescent markers expressed from a seed-specific promoter that flanks *CEN3*[48]. The red and green T-DNA insertions span a region of approximately 8.91 Mb in the Col-CEN assembly of chromosome 3, which contains the centromeric satellite arrays in addition to flanking pericentromeric heterochromatin (Fig. 3a)[48]. Meiotic crossover recombination events that occur within this region result in progeny inheriting either red or green fluorescent markers, the frequency of which can be measured by automated seed imaging[52]. We transformed *CTL3.9* F$_1$ double hemizygous plants with the SunTag-SDG2:LRCen3_g construct (Fig. 3a, b). The F$_2$ plants represented individual transformants of the SunTag (T$_1$ for SunTag), and so F$_3$ seeds from individual F$_2$ plants were imaged to measure the rate of crossover recombination, in centiMorgans, over the centromere 3 spanning region of *CTL3.9*[48].

Remarkably, the SunTag:SDG2:LRCen3_g containing lines showed significantly elevated crossover recombination within the *CTL3.9* interval as compared to the non-SunTag control lines (ANOVA with post-hoc Tukey HSD cutoff <0.05) (Fig. 3c, Supplementary Fig. 9). Some individual transgenic plants displayed a substantial increase in *CTL3.9* crossover recombination rate, with over 50% higher frequency, as compared to the average in non-transgenics, across this megabase-scale centromere-spanning region (Fig. 3c). SunTag:SDG2:LRCen3_g progeny from the most highly recombining line that also showed Mendelian inheritance of the *CTL3.9* T-DNAs was sown to assess whether the effect could persist in the subsequent (F$_4$) generation. The plants retained a significantly elevated crossover frequency across *CTL3.9* in the next generation, suggesting that the SunTag:SDG2:LRCen3_g effect is relatively stable in the presence of the transgene (Fig. 3c).

To examine the distribution of SunTag:SDG2:LRCen3_g in the high recombination lines (P4, Supplementary Fig. 9), we performed ChIP-seq (anti-HA) comparing progeny from sibling plants that either inherited (+SunTag), or did not inherit (-SunTag) the SunTag machinery, as well as non-transgenic (Col-0) controls. Chromosome-wide plots at 100 kb resolution showed that SunTag-SDG2 is enriched over the entire centromeric region of chromosome 3 (Fig. 3d). SunTag was also enriched over the other four *A. thaliana* centromeres (Supplementary Fig. 10). As these regions are highly repetitive, we reasoned that the LRCen3 guide RNA might additionally bind the other centromeres due to mismatch tolerance. Indeed, while the LRCen3 guide RNA aligns perfectly to 241 loci within the centromere of chromosome 3, by allowing a single mismatch, the guide is predicted to bind to target sites in all 5 centromeres (Supplementary Fig. 11), which is consistent with the ChIP-seq data observed. Having confirmed that SunTag is enriched within the centromeric regions in these lines, we next examined levels of H3K4me3. Inspecting centromeric regions, we observed many instances of novel or elevated H3K4me3 peaks in the LRCen3 targeted lines (Supplementary Fig. 12). Quantifying H3K4me3 levels at these peaks confirmed the enrichment of H3K4me3 levels over the centromere region and further extending into the neighbouring pericentromeres (Fig. 3d, Supplementary Fig. 13). As the markers on *CTL3.9* span the centromere and adjacent pericentromeric regions[48] (Fig. 3d), this is consistent with the significantly elevated

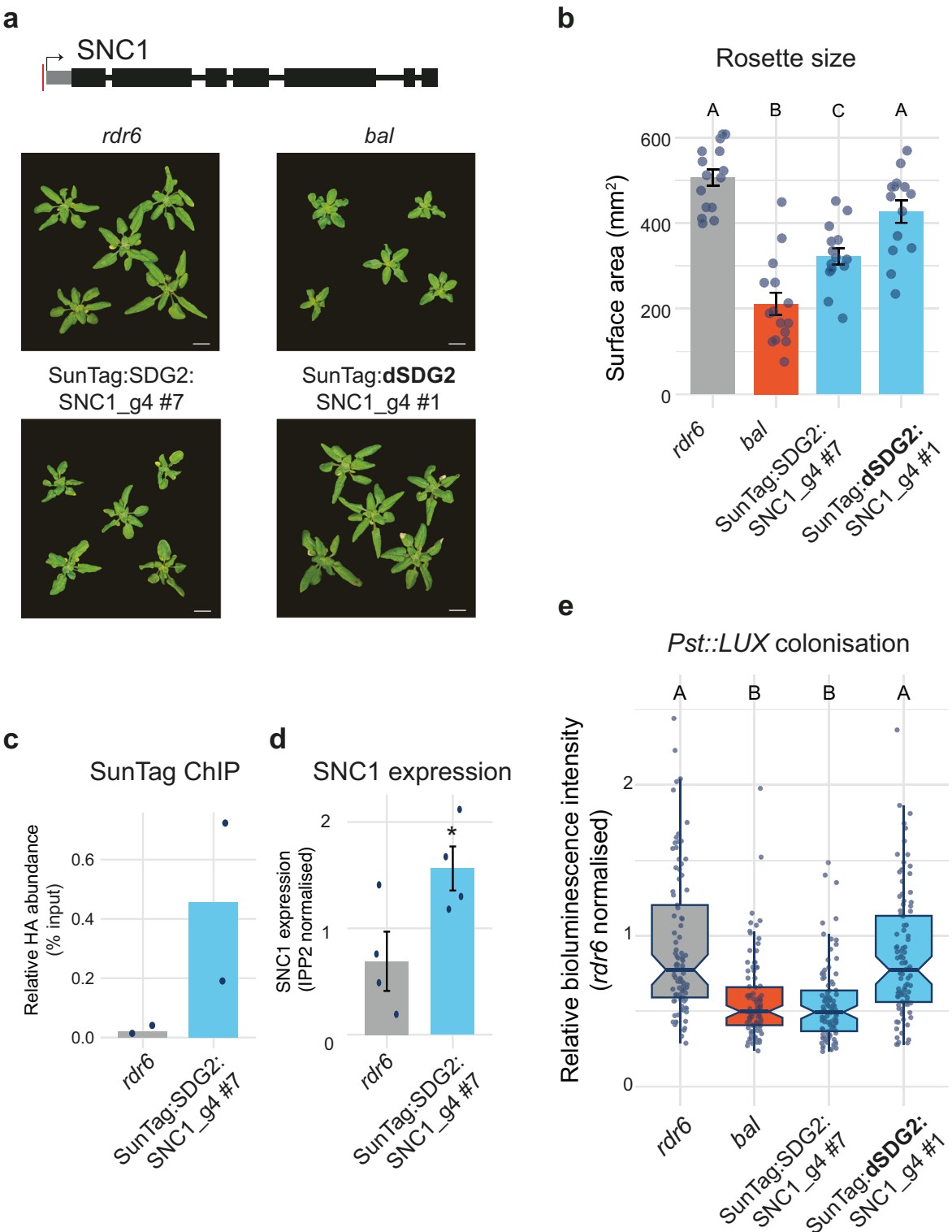

**Fig. 2 | SunTag-SDG2 targeting to *SNC1* enhances resistance to *P. syringae.*** **a** Upper, schematic of the *SNC1* gene, with the red line indicating the *SNC1* gRNA target site, drawn to scale. Lower, representative images of 3-week-old plants from the genotypes indicated. Scale bar = 1 cm. **b** Rosette size quantification. Each dot represents an individual plant. Error bars represent SEM. Different letters indicate significant difference by ANOVA with post-hoc Tukey HSD, $p < 0.05$. $n = 14$ for SunTag:dSDG2:SNC1_g4 #1, $n = 15$ for the rest. **c** ChIP qPCR for the presence of SunTag at SNC1. **d** RT-qPCR for *SNC1* expression. Error bars represent SEM. * indicates *p*-value = 0.045 by Student's *t*-test. $n = 4$ biological replicates (each a pool of 10 seedlings). **e** *Pst::LUX* assay for colonisation quantification at 3 days post-inoculation. Different letters indicate significant difference by ANOVA with post-hoc Tukey HSD, $p < 0.05$. Boxplots show the median, the interquartile range, whiskers extending to 1.5× the interquartile range, and individual data points plotted as dots. $n = 96$ wells, containing 3 seedlings each.

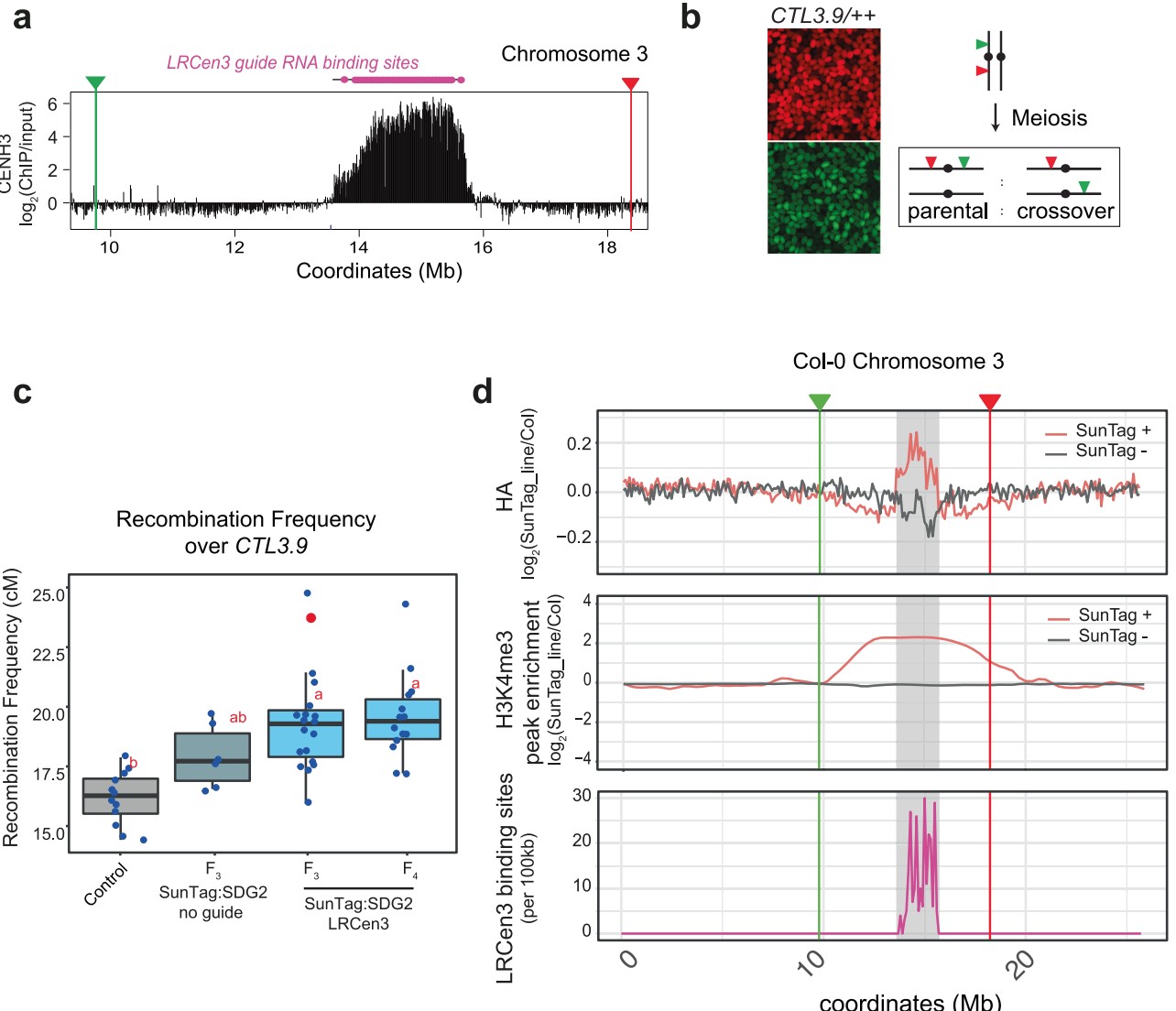

**Fig. 3 | Centromeric targeting of SunTag-SDG2 elevates meiotic crossover recombination rate. a** Region of chromosome 3, showing CENH3 ChIP-seq enrichment[50] with *CTL3.9*[51] red/green T-DNA marker positions and LRCen3 guide RNA binding sites indicated. **b** Left panel, representative image of red and green fluorescing seeds from a *CTL3.9* double homozygous line. Right panel, schematic showing how crossovers are identified within *CTL3.9*. **c** Boxplot showing crossover recombination frequency over *CTL3.9* in centiMorgans. Control data is from Col-0 (non-transgenic) crossed to *CTL3.9*. Different letters indicate significant difference by ANOVA with post-hoc Tukey HSD ($p < 0.05$). Control: $n = 12$, LRCen3 F3: $n = 19$, No guide: $n = 6$, LRCen3 F4: $n = 14$. Boxplots show the median, the interquartile range, whiskers extending to 1.5× the interquartile range, and individual data points plotted as dots. The red dot indicates the seed set line taken to the next generation (F$_4$). **d** Chromosome-wide plots of chromosome 3. Upper panels show enrichment of SunTag (anti-HA) by ChIP-seq in sibling lines with/without SunTag (±). Enrichment is calculated as log$_2$ fold change over non-transgenic (Col-0) controls in 100 kb windows. The middle panel shows H3K4me3 enrichment over H3K4me3 peaks (log$_2$ fold change over non-transgenic (Col-0) controls). The lower panel depicts an LRCen3 guide RNA binding site density over chromosome 3. *CTL3.9* marker positions are indicated, and centromeric regions are shown in grey. All data are mapped to the Col-CEN genome assembly[47].

crossover recombination rate observed. Overall, the results demonstrate that centromerically targeted SunTag-SDG2 can elevate the meiotic crossover potential of these typically recombination-suppressed regions.

**Mammalian PRDM9 drives efficient transcriptional activation in Arabidopsis**

As the enrichment of H3K4me3 in SunTag:SDG2:LRCen3_g was not exclusive to the guide RNA-targeted centromeric regions, we reasoned that the SunTag system may afford some level of off-target activity due to non-guide RNA-directed ectopic interactions. Consistent with this observation, we also noticed that our SunTag-SDG2 no-guide RNA controls frequently exhibited pleiotropic developmental phenotypes (Supplementary Fig. 14). To quantify global levels of H3K4me3, we

performed bulk histone westerns, comparing no-guide RNA lines to *FWA* expression-activating guide RNA-containing lines. Importantly, the SDG2 effector itself is expressed to similar levels as in the *FWA* targeting lines (Supplementary Fig. S3a). The SunTag-SDG2 no guide lines all showed significant (3–4 fold) increases in global levels of H3K4me3, as compared to the *FWA* guide containing lines (Supplementary Fig. 15A). This indicates that the presence of the guide RNA may enhance the targeting specificity of the SunTag complex by reducing the frequency of non-specific interactions that result in off-target deposition of H3K4me3. We also observed that a small number of SunTag:dSDG2:FWA_g4 lines could activate *FWA* expression, but this effect was limited to lines where the dSDG2 effector was highly overexpressed (Supplementary Fig. 15B). To systematically examine this effect, we screened T$_1$ plants finding that 6% (1/16) of the *FWA*

targeted dSDG2, and 14% (2/16) of the no-guide SDG2 lines could activate *FWA* expression (Supplementary Fig. 16). This underscores the potential off-target effects from SunTag-SDG2, and suggests that dSDG2 either retains some residual catalytic activity, or that it can recruit endogenous machinery to initiate transcription independently of H3K4me3 deposition.

In order to reduce off-target activity and the potential for co-recruitment of endogenous complexes, we replaced SDG2 with an orthogonal effector, PRDM9. PRDM9 is a mammalian H3K4me3 methyltransferase, and Policarpi et al.[31] recently showed that the catalytic domain could be incorporated into the SunTag system for targeting of H3K4me3 in mammalian cells. Therefore, we inserted the PRDM9 methyltransferase domain, as well as a catalytically deactivated version, into our SunTag system and targeted *FWA* (Fig. 4a). Strikingly, SunTag:PRDM9:FWA_g4 was highly effective at activating *FWA* mRNA expression (Fig. 4b). In the $T_1$ generation, 72% (23/32) of SunTag:PRDM9:FWA_g4 plants could activate FWA, while none of plants with the dPRDM9 containing (0/16), or no-guide RNA (0/19) constructs were able to do so (Supplementary Fig. 17). Importantly, we could detect robust expression of the effector from all constructs (Supplementary Fig. 17). ChIP qPCR confirmed the presence of SunTag and the enrichment of H3K4me3 at *FWA* (Fig. 4c). At the transcriptome level, the effect of SunTag:PRDM9:FWA_g4 was highly specific, with only 19 genes identified as differentially expressed (15 up, 4 down, FDR < 0.01), and with *FWA* itself having the highest fold change amongst them (>158 fold change over *rdr6*) (Fig. 4d). Unlike the no-guide SDG2 transgenic lines, bulk histone westerns confirmed no H3K4me3 increases in the PRDM9 no-guide lines (Supplementary Fig. 18). Correspondingly we did not observe any developmental defects and saw only minor transcriptional defects when comparing the no-guide lines to non-transformed *rdr6* controls (Supplementary Fig. 19). Finally, we performed ChIP-seq in these lines, confirming *FWA* as the most significant and enriched peak for SunTag binding and revealing deposition of H3K4me3 at the transcriptional start site of *FWA* (Fig. 4e).

At the genome-wide level, while SunTag:PRDM9:FWA_g4 lines also displayed a modest decrease in H3K4me3 signal at endogenous peaks with a corresponding increase over 3′ gene body regions (Fig. 4f, Supplementary Fig. 20), these effects were less pronounced than SunTag-SDG2 (Fig. 1g), again suggesting that the PRDM9 effector fusion has reduced off-target potential. Finally, we performed the *CTL3.9* meiotic recombination assay, finding that PRDM9 was similarly effective at increasing crossover frequency over the centromeric *CTL3.9* interval (Fig. 4g, Supplementary Fig. 21). Overall, our results demonstrate that PRDM9 is highly effective for deposition of H3K4me3 in plant, as well as mammalian SunTag systems. Furthermore, as PRDM9 is not likely to have endogenous partners in the plant nucleus, the results provide strong evidence in favour of a direct role for H3K4me3 in transcriptional stimulation.

## Discussion

Our results demonstrate that targeted deposition of H3K4me3 using the SunTag system enables transcriptional activation, enhanced pathogen resistance, and increased crossover recombination over megabase-scale centromere-proximal regions. These findings support an instructive role for H3K4me3 in transcriptional control and open new avenues for precise, potentially non-GM crop improvement through epigenome engineering. Because the effect depends on the presence of a functional catalytic domain and is observed using both native and non-native methyltransferases, it is unlikely to result from co-recruitment of endogenous complexes, strengthening the argument that H3K4me3 itself can act as a causal regulatory mark.

In a related study, Wang et al.[38] showed that full activation of *FWA* by recruitment of SDG2 required active removal of DNA methylation by ROS1 and related proteins. Whether H3K4me3 directly recruits

ROS1 machinery is currently unknown. However, the finding that PRDM9 also causes potent transcriptional activation of *FWA* suggests that ROS1 recruitment is downstream of H3K4me3, rather than being co-recruited via COMPASS complex components[9,10,53]. Removal of DNA methylation could also help explain the increased centromeric crossover recombination observed over the *CTL3.9* interval, consistent with prior work showing elevated recombination rates in non-CG DNA methylation mutants in a similar interval[48,54]. However, in mammalian systems, H3K4me3 has been shown to stimulate transcription by promoting Pol II pause release[8]. Similarly, using zinc finger-fused SDG2 in *A. thaliana*, Wang et al.[38] found many instances of ectopic H3K4me3 driving transcriptional stimulation in the absence of pre-existing DNA methylation or DNA methylation removal. Thus, DNA methylation-independent mechanisms must also play a role in H3K4me3-mediated transcriptional activation.

In our system, SunTag-SDG2 and PRDM9 targeting of *FWA* with a single guide caused strong upregulation at this locus, yet we did not observe any plants that switched to late-flowering as previously reported when *FWA* transitions to an active *fwa* epistate. The Wang et al.[38] study only observed flowering delays when two guides were used to target the *FWA* promoter. We cannot exclude the possibility that *FWA* failed to reach a critical expression threshold in our experiments, or that expression was developmentally mis-timed or mis-localised. It is also possible that in the single guide system, the H3K4me3 deposition induces more non-functional transcripts, consistent with chromatin's roles in co-transcriptional regulation[55,56]. The lack of a flowering time phenotype also aligns with the absence of transgenerational effects, suggesting that the targeting did not remove sufficient DNA methylation to establish heritable gene *FWA* expression. This is further supported by the modest H3K4me3 peak observed at *FWA* in our targeted lines, compared to lines that have fully switched to late flowering[38].

Both mammalian PRDM9 and the endogenous SDG2 catalytic domains were able to transcriptionally activate *FWA* and elevate crossover recombination rates over a recombination-suppressed centromere-spanning interval. Seminal work in *Drosophila melanogaster* showed that while loss of the H3K4 monomethyltransferases, *Trr*, was lethal, the corresponding catalytic-null caused only minor developmental defects[57]. This underscores the importance of using catalytic mutations and orthogonal effectors to separate the effects from histone marks themselves, over the histone-modifying enzymes and associated complexes. It also highlights the utility of modular targeting platforms that can decouple chromatin modifications from native pathways. Continued development of such tools will be crucial for both basic discovery and applied crop engineering[23,25,27,28,58–66].

We observed off-target effects in both SunTag-PRDM9 and SunTag-SDG2 transgenic lines, manifested as increased non-specific H3K4me3 background signal and reduced occupancy at endogenous locations (Figs. 1g, 4f, Supplementary Fig. 13). This off-target signal is inherently difficult to diagnose, as it is virtually indistinguishable from a ChIP-seq with reduced signal-to-noise. However, as we observed the same trend in multiple independent experiments, with the SunTag lines always showing this same pattern as compared to the non-transgenics (Figs. 1g, 4f, Supplementary Fig. 13A), we conclude that the effect is due to SunTag off-targeting, rather than ChIP signal variability. Notably, these effects were weaker in PRDM9 lines, supporting the idea that orthogonality reduces off-target potential. Unlike SDG2, PRDM9 lines without guide RNAs did not show elevated bulk H3K4me3 levels (Supplementary Fig. 15 vs. 18). The strong developmental phenotypes in the SDG2 no-guide lines (Supplementary Fig. S14) may result from transcriptional misregulation at ectopically modified loci, or could be caused by transcription-independent metabolic defects[5]. Interestingly, these phenotypes were largely absent in the SDG2 guide RNA-targeted lines, suggesting that the guide RNAs reduce this off-target activity. With the guided lines, it is worth reiterating that the SunTag on-target

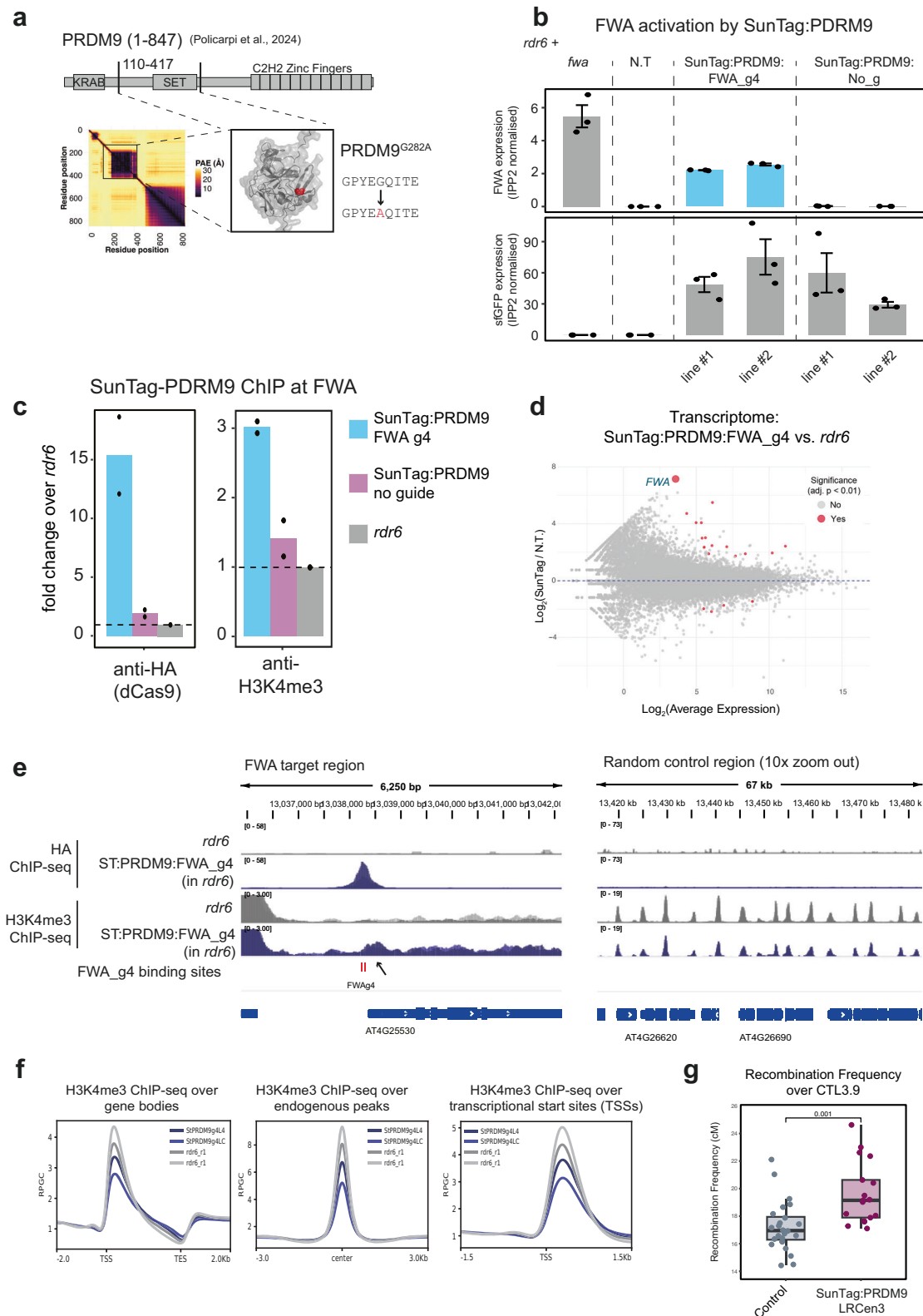

activity is highly specific for both SDG2 and PRDM9, with *FWA* as the most significantly enriched SunTag peak by ChIP-seq, and is the most upregulated transcript with few other DEGs observed (Figs. 1f and 4d). However, off-target effects and metabolic impacts are important considerations when designing these systems for use in crops, which possess larger genomes and more complex regulatory architectures than *A. thaliana*[67]. Notably, SunTag has already been successfully

deployed in rice for precise removal of DNA methylation in the context of cold stress adaptation[60], demonstrating the feasibility and potential of these epigenome engineering approaches in agricultural settings.

In summary, our work demonstrates that targeted deposition of H3K4me3 is a powerful tool for modulating transcription and meiotic crossover recombination in plants. By deploying both endogenous and orthogonal methyltransferases, we reveal core insights into

**Fig. 4 | SunTag-PRDM9 is sufficient for activation of *FWA*. a** Upper panel shows PRDM9 with catalytic SET domain (drawn approximately to scale). The lower left depicts a predicted alignment error (PAE) plot from AlphaFold3 prediction of the PRDM9 coding sequence with a box drawn around the region cloned into SunTag (amino acids 110–417). The lower right depicts the AlphaFold3 model for PRDM9$_{110-417}$, with the amino acid change for dPRDM9 indicated (position shown in red). **b** RT-qPCR for *FWA* (upper panel) and the effector module (sfGFP, lower panel) for the genotypes indicated. Dots represent individual plants, with two independent T$_3$ lines used per construct, 3 biological replicates per construct. Error bars represent SEM. **c** ChIP qPCR for the presence of SunTag (left panel) and H3K4me3 enrichment (right panel) over the TSS of *FWA*. **d** MA plot comparing the transcriptome of SunTag:PRDM9:FWA_g4 as compared to the non-transformed control (*rdr6*). Differentially expressed genes were defined using an adjusted *p*-value (Bonferroni method), shown in red (FDR < 0.01), with FWA labelled and enlarged for visibility. **e** Genome browser image showing efficient SunTag targeting and H3K4me3 enrichment at *FWA* and at a random genic control region. Biological replicate tracks from independent lines are overlaid. **f** H3K4me3 ChIP-seq metaplots over all genes, H3K4me3 endogenous peaks, and TSS regions. **g** Boxplot showing crossover recombination frequency over *CTL3.9* in centiMorgans (*p* < 0.001, two-sample, two-sided *t*-test). Boxplots show the median, the interquartile range, whiskers extending to 1.5× the interquartile range, and individual data points plotted as dots. *n* = 24 for control, *n* = 17 for Suntag:PRDM9:LRCen3_g.

H3K4me3 function and highlight the potential of this platform for translational application. Future optimisation of guide design, effector choice, and off-target mitigation will be important for enabling both mechanistic discovery and crop engineering through epigenome editing.

## Methods

### Plant materials and growth conditions
All *Arabidopsis thaliana* lines are in a Col-0 ecotype background. Plants grown on plates were sown onto 1/2× Murashige–Skoog (MS) medium, 1% sucrose, and 0.8% agar (pH 5.7), stratified for 2 d at 4 °C in the dark, then transferred to growth chambers (Percival CU-41L4D) at 21 °C under long day light conditions (16 h light/8 h dark). Plants grown on soil were grown in F2 soil at 20 °C under long day light conditions (16 h light/8 h dark). Lines: *rdr6-15* (SAIL_617_H07), and *fwa* lines were previously described[27]. NahG and *edr1* were kindly provided by Prof. Juriaan Ton, previously described in ref. [45]. *bal* lines were previously described in ref. [41]. The CTL 3.9 line was previously described in refs. [48,51]. All transgenic plants were generated by *Agrobacterium tumefaciens* strain AGL1 using floral dip[68].

### Plasmid construction
To generate the SunTag:SDG2:FWA_g4 construct, the CDS of the catalytic domain of SDG2 (from amino acids 1571–2335) was amplified and cloned into the BsiWI linearised SunTag construct (previously described[28]), using In-Fusion (Takara). To generate dSDG2: the conserved amino acid predicted to abrogate catalytic activity when mutated (Y1903F) was identified using ConSurf[69]. Overlapping PCR was used to generate a version of SDG2 containing the desired mutation (see Supplementary Table 1), and the amplified PCR product was cloned into a BsiWI linearised SunTag-SDG2 construct by In-Fusion (Takara). To generate SunTag-PRDM9, the catalytic domain of mouse PRDM9 (110-417aa) (described in Policarpi et al.[31]) was synthesised (TWIST Bioscience) and cloned into the SunTag using the same BsiWI linearization and In-Fusion method described above. Subsequently, overlap PCR was used to generate a version of PRDM9 with the G282A mutation (see Supplementary Table 1). The SNC1 (guides were chosen based on the predicted binding proximity to the TSS), LRCen3, and the no guides controls were cloned using the protocol described in ref. [70] (see Supplementary Table 1).

### Chromatin immunoprecipitation
The chromatin immunoprecipitation experiments were performed as previously described[71]. 2 g of 14-day-old seedlings were collected per biological replicate. For SunTag-SDG2 FWA and SunTag-SDG2 SNC1, the input material is pooled T$_2$ plants. For the SunTag:PRDM9:FWA_g4 in *rdr6*, 2 independent T$_3$ lines expressing FWA were used alongside *rdr6* controls. 7 μl of anti-HA (3F10, Merck) and 5 μl of anti-H3K4me3 (ab8580, Abcam) were added to up to 2 ml of sheared chromatin per sample for pulldown. The ChIP-purified DNA was directly used for ChIP qPCR and ChIP-seq library preparation. Primer information is listed in Supplementary Table 1. Values were normalised to Input

($2^{-(ChIP\ Ct\ -\ Input\ Ct)}$*100). ChIP-seq libraries were generated using NuGen Ovation Ultra Low System V2 kits according to the manufacturer's instructions and were sequenced on an Illumina NovaSeq X instrument with 150 bp paired-end reads.

### Crossover measurement
For the recombination rate experiment, SunTag plasmids were transformed into F$_1$ plants derived from a cross between the traffic line *CTL3.9* (containing insertions of NAP:eGFP and NAP:dsRED) and Col-0 (Wu et al.[51]). Note that as the red and green T-DNA insertions are separated by nearly 9-Mb, disruption from the T-DNA insertions (5-Kb and 12-Kb, respectively) is likely to be negligible at this scale[48]. To help reduce the impact of transgene silencing in the absence of the *rdr6* background (due to the impracticality of generating homozygous *rdr6* in the double heterozygous *CTL3.9* F1 background) and to ensure a sufficient number of scorable lines, we aimed to screen 70–100 T$_1$s per construct. T$_1$ transgenic plants (F$_2$ plants) were recovered on hygromycin selection plates. Seeds harvested from T$_1$ (F$_3$) plants were first pre-screened using a Leica DFC310 FX dissecting microscope with ultraviolet filters. Only seeds that contained both red and green markers were cleaned to remove plant debris and included in the analysis pipeline. The seed monolayer containing between 1000 and 2500 seeds was captured with a Leica DFC310 FX dissecting microscope, first using brightfield, followed by UV through a dsRED filter and then UV through a GFP filter. Images were analysed using a modified CellProfiler pipeline (version 4.2.5) described in refs. [48,52,72]. For the F$_4$ data, seeds derived from the most highly recombining and non-distorted line (P4) were sown (F$_3$), grown for a generation, and seeds from individual plants were collected (F$_4$) for recombination rate analysis as above, again filtering seed sets from which either of the fluorescent markers had become homozygous (presence/absence) in the previous generation.

### Western blots
Western blots were performed as previously described[73] with minor modifications described below. For the detection of the large dCAS9-10xGCN4 and SDG2 effector components, as shown in Fig. S1, single leaf punch (1 cm diameter) samples from individual plants were used as input. Tissue was ground in a Tissuelyser II (Retsch) and resuspended in 100 μL of a 60:40 ratio of 2XSDS:8 M Urea, mixed, and boiled at 95 °C for 5 min. Samples were centrifuged at >10,000 × *g* to remove debris and loaded into a 3–8% tris acetate gel using MOPS running buffer. After electrophoresis, the protein was transferred from the gel to a methanol-activated PVDF membrane in Tobin Buffer (25 mM Tris, 192 mM Glycine, 20% methanol, 0.035% SDS). Membranes were blocked in 5% milk (w/v) in PBST (Phosphate Buffered Saline, 0.1% Tween20) and then incubated with HRP-conjugated anti-HA at a 1:3000 dilution in block. The blot was imaged by ECL detection.

For the western blots used to detect levels of H3K4me3, 100 mg of 7-day-old seedlings were frozen in liquid nitrogen and ground using a Tissuelyser II (Retsch). Total histones were extracted using the EpiQuik™ Total Histone Extraction Kit (EpigenTek) according to

the manufacturer's instructions. 5 µl of sample for H3 and 20 µl of sample for H3K4me3 were loaded onto a NuPage 4–12% BIS-Tris gel (Invitrogen) in NuPage MOPS SDS running buffer (Invitrogen). Transfer from gel to PVDF membrane was performed using an iBlot 3 machine with the dedicated transfer stack (Invitrogen). Following transfer, the membrane was blocked in 5% milk (w/v) in TBST (Tris Buffered Saline, 0.1% Tween20) and shaken at room temp for 1 h. Blocking buffer was discarded, and membranes were incubated in primary antibody (anti-H3, ab1791 and anti-H3K4me3, ab8580, Abcam) in blocking buffer (1:1000), shaking overnight at 4 °C. The following morning membrane was rinsed in TBST 5 times for 5 min at room temperature. Secondary antibody was added for 1 h (Goat anti-rabbit HRP, G-21234, Thermo). The membrane was washed again 5 times for 5 min. Detection was performed using a SuperSignal West Pico chemiluminescent substrate (#34080, Thermo) using an ImageQuant 800 (Amersham).

### Fluorescent imaging
Roots of 7-day-old plants grown in vertical 1/2× MS solid media plates were imaged using a Leica DM6000B epifluorescent microscope.

### Rosette size surface area measurement
3-week-old plants grown on soil were imaged using a Google Pixel 7 mounted on a tripod. Images were then analysed using Fiji.

### *Pseudomonas syringae* colonisation assay
Infection assays were performed with *P. syringae* pathovar *tomato* strain DC3000 (*Pst*) strain expressing a stable chromosomal insertion of the *lux-CDABE* operon from *Photorhabdus luminescens* (*Pst::LUX*), using a slightly modified version of a previously published protocol (Furci et al.[45]). Plants were grown on 1/2× MS solid media supplemented with 200 µg/ml Timentin in 96-well plates (Falcon) for 7 days. Prior to inoculation, *Pst::LUX* were grown in King's B media containing 50 µg/ml Kanamycin and 50 µg/ml Rifampicin overnight at 28 °C. Bacteria were pelleted by centrifugation, washed in 10 mM $MgSO_4$, and finally resuspended to an OD600 of 0.2 in 10 mM $MgSO_4$ containing 0.015% v/v Silwet-L77. 7-day-old seedlings were sprayed with bacteria solution, and 96-well plates were sealed with parafilm to maintain 100% relative humidity. Seedlings were imaged at 1–4 days post-infection (dpi). Prior to imaging using (ImageQuant 800, Amersham), plates were left in the dark for 2 min. Bacterial bioluminescence images were acquired with 4 min exposure time, and brightfield images were taken using OD settings of the ImageQuant 800. Image-based quantification of bioluminescence was carried out in Fiji. For each well, the well outline was obtained from brightfield images and added to the ROI Manager. These ROIs were then transposed onto the bioluminescence images. The bioluminescence intensity from infected seedlings was obtained using the Fiji functions Analyse, Measure, and Mean value.

### RT-qPCR
RNA was extracted from the indicated plant material using TRIzol reagent (Invitrogen) and the Direct-Zol RNA MiniPrep (Zymo) kit, including in-column DNase treatment following the manufacturer's instructions. cDNA was synthesised using SuperScript IV (Invitrogen). RT-qPCRs were performed using a Luna Universal qPCR Master Mix (NEB) and a CFX connect Real-time PCR detection system (Bio-Rad). For SunTag-SDG2 targeting FWA and SNC1, RNA was extracted from 14-day-old whole seedlings grown on 1/2x MS plates. SNC1 targeted lines were treated with *Pst::LUX* 4 h prior to harvesting. For SunTag-PRDM9 RT-qPCR, RNA was extracted from 4-week-old leaf tissue of all $T_1$ lines and 2 independent $T_3$ lines per construct. 450 ng of total RNA was used for cDNA synthesis. Primers are listed in Supplementary Table 1.

### Quant-Seq library construction
For Suntag-SDG2 QuantSeq, 350 ng of total RNA from ~10 pooled $T_2$ plants were used as input material. For Suntag-PRDM9 Quant-seq, 350 ng of total RNA from 3 $T_1$ SunTag:PRDM9:FWA_g4 plants expressing FWA, 3 $T_1$ SunTag:PRDM9:No_g lines and *rdr6* and plants were used for library preparation. Libraries were prepared using QuantSeq 3' mRNA-Seq Library Prep Kit FWD (Lexogen) according to the manufacturer's instructions and sequenced on the Illumina NovaSeq 6000 PE150 instrument.

### Bioinformatic analysis
**ChIP-seq analysis.** The ChIP-seq analysis pipeline was performed as previously described with minor modifications. Bowtie2 (version 2.5.0) was used to map the PE150 bp read data in fastq format to the TAIR10 genome (--no-unal), and was converted to bam format using Samtools (version 1.10). Reads were de-duplicated using the samtools fixmate and markdup commands (see https://github.com/C-Jake-Harris/Binenbaum-SunTag-H3K4me3 for more details). Tracks were generated in DeepTools (version 3.5.1) using bamCoverage (--normalizeUsing RPGC, --effectiveGenomeSize 135000000 --binSize 10) with multicopy regions blacklisted (--blackListFileName) using the regions identified in ref. [74]. For analysis of reads mapping to the centromere, a modified version of the previously described pipeline[50] was used. Briefly, PE150 bp read data in fastq format were mapped to the Col-CEN genome[47] using Bowtie2 (version 2.2.5) with (--very-sensitive -k 200 --no-unal --no-discordant) and were filtered for primary alignments using samtools view (-F 256 -q 5) prior to deduplication and track generation, as above. Peaks were called using MACS2 (version 2.2.9.1) with default parameters. multiBigwigSummary (BED-file --outRawCounts) was used to gather normalised read enrichment over called peaks. Correlation plots were generated in DeepTools using MultiBamSummary (--binSize 25) and PlotCorrelation (-c pearson --removeOutliers --plotNumbers -p heatmap).

**QuantSeq analysis.** In accordance with the manufacturer's (Lexogen) instructions, only the read (*1.fq.gz) from PE150 bp data was used for downstream analysis. Briefly, reads were trimmed using cutadapt (version 1.18) and were mapped to the genome using STAR (version 2.7.10b) to the TAIR10 genome (--quantMode GeneCounts --alignIntronMax 10000 --outSAMmultNmax 20). Gene counts were used as input for DESeq2 analysis in R. Differentially expressed genes were defined using an adjusted *p*-value (Bonferroni method) cutoff of <0.01.

**gRNA binding analysis.** For the LRCen3 binding and mismatch analysis, the LRCen3 guide RNA sequence (5'-AGGCTTACAAGATTGGGTTG-3') was mapped to the Col-CEN genome using bowtie2 (version 2.2.5). The following options were used for no mismatches (-f -a --end-to-end --np 0 --score-min L,0,0) and for 1 mismatch (-f -a --end-to-end --score-min L,0,−1). Mapped reads were sorted and converted to bam format using Samtools (version 1.10), and subsequently converted to .bed format using the bamtobed function in bedtools (2.20.1). Bed files were used for downstream analysis in R to generate chromosome-wide density plots over 100 kb regions.

**Statistics and reproducibility.** Statistical significance was determined by using an unpaired two-tailed Student's *t*-test for two-group comparisons and ANOVA with post-hoc Tukey HSD for multiple comparisons. Asterisks indicate significant differences (*$P < 0.05$, **$P < 0.01$, ***$P < 0.001$). Different lowercase letters indicate significant differences ($P < 0.05$). No statistical method was used to predetermine sample size, no data were excluded from the analyses, the experiments were not randomised, and the investigators were not blinded to allocation during experiments and outcome assessment.

## Reporting summary

Further information on research design is available in the Nature Portfolio Reporting Summary linked to this article.

## Data availability

The high-throughput sequencing data generated has been deposited in NCBI (GSE288686). Previously published datasets: CENH3 ChIP-seq and input are on ArrayExpress (E-MTAB-11974)[47,50]. Custom scripts used are available on GitHub (https://github.com/C-Jake-Harris/Binenbaum-SunTag-H3K4me3). Source data are provided with this paper.

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

## Acknowledgements

We would like to thank Jurriaan Ton's group at the University of Sheffield for their generous gift of the *Pst::LUX* strain and for their support in helping to establish the infection and quantification protocol in the Harris group. C.J.H. is supported by a Royal Society University Research Fellowship (URF\R1\201016) and an ERC Starting Grant (TransPlantMemory), funded by UKRI EPSRC (EP/X025306/1). J.B. is supported by a Marie Curie postdoctoral fellowship, funded by UKRI (EP/Z001749/1). V.A. is supported by a 4-year BBSRC-funded doctoral studentship. I.R.H. is supported by UKRI/BBSRC grants BB/Y009487/1 and BB/V003984/1, and an ERC Advanced Grant (EvoPanCen). N.G. is supported by a BBSRC-UKRI doctoral studentship. S.E.J. is an investigator of the Howard Hughes Medical Institute.

## Author contributions

C.J.H., I.R.H. and S.E.J. conceived, designed and supervised the research. C.J.H. performed experiments, NGS bioinformatic analysis, and assembled the manuscript. J.B. performed the majority of the experiments relating to SunTag-SDG2. V.A. performed the experiments relating to SunTag-PRDM9. H.F. performed the majority of experiments and analysis relating to meiotic recombination. L.X. performed the SunTag-SDG2 centromere-targeting ChIP-seq. N.G. provided the Col-0

x *CTL3.9* F$_1$ seeds and guidance on the analysis of recombination frequency. P.W. and R.B. contributed to the design of the centromere-targeting guide RNA. A.P. cloned SDG2 into SunTag.

## Competing interests

The authors declare no competing interests.
