## [Transparent Peer Review file · Nature Communications]

CRISPR targeting of H3K4me3 activates gene expression and unlocks centromere-proximal crossover recombination in Arabidopsis

Corresponding Author: Professor C. Jake Harris

Version 0:

Reviewer comments:

Reviewer #1

(Remarks to the Author)

I read the manuscript of Binenbaum et al. with great interest. In this is a very carefully done study. The authors were able to show that SunTag-mediated targeting of H3K4me3 can influence different biological processes in Arabidopsis. They could show an enhancement of gene expression, which in their readout led to an increase in pathogen resistance. Even more exciting, they were able to change meiotic recombination patterns by the use of a chromosome 3 centromere repeat specific sgRNA. Although the effect was minor and the specificity due to the presence of many similar repeats limited, the fact that the authors were able to redirect recombination by a CRISPR/Cas based approach is notable. The results are novel, represent a valid proof of concept and of interest to a wide audience.

Reviewer #2

(Remarks to the Author)

Binenbaum and Adamkova et al present epigenome targeting of H3K4me3 using the plant SDG2 and the mammalian PRDM9 histone methyltransferases. They convincingly show that the SunTag system rather specifically targets these histone methyltransferases to their targets to activate gene expression. This was shown by targeting FWA and also SNC1 in Arabidopsis. It's fascinating that the dSDG2 can also reactivate FWA at a low frequency and it seems to me that PRDM9 might be able to as well (see comments below). I think this is a part of the story that is undersold and quite intriguing. This study also revealed increased frequency of recombination by targeting H3K4me3 to the centromeres, which is an important result given these regions of the genomes are somewhat recalcitrant to recombination in many plant species.

Overall, this is a really nice study that will pave the path forward to epigenome editing and improved recombination in plants. The comments below are intended to improve this already strong manuscript.

1. SunTag efficiently targets H3K4me3 to FWA in a gene specific manner. This result would be strengthened by adding ChIP-seq of H3K4me3 instead of PCR.
2. Figure 3d. Enrichment of CEN targeting SDG2 looks great, but the H3K4me3 ChIP is not as convincing.
 - a. What is the quality of the ChIP-seq data? There is a lot more H3K3me3 in the euchromatin and centromere making this hard to interpret. Is it possible the ChIP did not work or is there off targeting of H3K4me3?.
 - b. The replicate data in the supplement show a convincing results. I would add some standard metaplots of H3K4me3 around the TSS to show the quality of the ChIP data.
 - c. How do you reconcile the specificity in Figure 1 with the widespread off-targeting in Figure 3?
 - d. Where is the ectopic H3K4me3 located? Is it in gene bodies or TSSs. A closer inspection would be interesting to know for the readership.
3. The PRDM9 targeting of H3K4me3 is quite widespread throughout the genome body raising the possibility that it is either off-targeting or H3K4me3 is weakly enriched. Is it possible that PRDM9 can also initiate activation by recruiting additional factors similarly to SDG2?
4. Data should be deposited to GEO.
5. Please add better labels to the samples in Figure S7 as it's unclear what they currently refer to.

Reviewer #3

(Remarks to the Author)

In this study, Binenbaum et al. investigate the causal role of H3K4me3 in regulating transcription and meiotic recombination. By employing a CRISPR-based SunTag system guided by single guide RNAs, they introduce H3K4me3 marks at specific genomic loci using two histone modifiers, SDG2 and the mammalian PRDM9. Their results show that directing H3K4me3 deposition with SDG2 to the previously silenced FWA locus successfully activates transcription. They also demonstrate that targeting the SNC1 disease resistance gene with SDG2 enhances its expression, thereby increasing Arabidopsis resistance to *Pseudomonas syringae*. Beyond gene activation, the authors show that applying H3K4me3 marks to centromeric regions, typically associated with very low recombination rates improves crossover frequency. Finally, by using PRDM9 as the H3K4me3 writer, off-target effects are reduced and the overall efficacy of the SunTag system is increased. While these findings present new tools for achieving precise epigenetic modifications in plants, important questions remain to be addressed.

Major Points:

1. It is important to show a strong causal relationship between increased H3K4me3 marks and the increased gene expression & phenotype. However, the current conclusion is compromised by 1) wide-spread off-target H3K4me3 deposition on a genome-wide scale, and 2) the use of only one or two selected lines to represent each construct (the gold standard is to use three independent transgenic lines). So, it is important to share data for all recovered lines to better understand the variability and efficiency of the system, and hence carefully draw conclusions from there. Since, in the methods section, the authors discuss that they have performed RT-qPCR on recovered 14-day-old whole seedlings with SDG2 experiments and all T1 lines with PRDM9. I recommend including a supplementary table or figure summarizing data from those transgenic lines for expression of the genes FWA and SNC1.
2. Since the authors claimed that PRDM9 is better than SDG2, it is necessary to have more comparisons of these two systems, which is needed to support the author's claim. Spending the majority of the work only on SDG2, not the comparison of these two, does not help make this conclusion.
3. Is this epigenome stable across generations? Does the epigenome editing depend on the presence of transgene? These questions need to be answered in this manuscript.
4. While the authors have demonstrated multiple use case scenarios of H3K4me3 depositions in Arabidopsis. Evidence shows low activation levels as demonstrated by qPCR and sequencing results. Since Arabidopsis has a small genome, this would raise a question to the applicability in crops, given their more complex genome and difficulty in transformations. Authors should at least discuss considerations and potential challenges while applying these tools/methods in higher plants other than Arabidopsis.
5. For centromeric targeting of SDG2. Col-0 plants were crossed with CTL3.9 to get CTL3.9 F1 double hemizygous. This F1 was transformed with SunTag:SDG2 constructs. I am confused to why the authors are now leaving behind *rdr6* mutants. Initially the authors have benchmarked *rdr6* as the best for the expression of SunTag and used it in multiple sections of the paper until centromeric targeting. Why?
6. In previous work, flowering time changes were observed because of FWA perturbations (DOI:10.1073/pnas.1716945115). Did H3K4me3 deposition, mediated by SDG2 and/or PRDM9, result in any observable phenotypes? Could you provide more details on these observations?

Minor Points:

1. The use of '#' annotations (e.g., #12) throughout the manuscript is unclear and inconsistent. For example, in Figure 1c, 'SunTag:SDG2:FWA_g4 #12' could be interpreted as a line number, but the legend says error bars represents three biological replicates, which would mean #12 is not a line number. Clarify the meaning of '#' annotations across all figures and text.
2. For Figure 3c, the control should have been CTL3.9 F1 double hemizygous lines crossings with no activation instead of Col-0 crossed with CTL3.9. Even though this control is provided in the supplementary material, it should also be included in Figure 3c for clarity and better representation. This is because CTL3.9 F1 double hemizygous lines form the background used to improve recombination frequency and already have a partially disturbed centromeric region of chromosome 3 due to the GFP-dsRed insertions. Recombination with Col-0 plants would be biased, as their centromere and pericentromeric regions are not disturbed, and they naturally exhibit low recombination rates.
3. L-40, grammar correction "its"(not "their") performance.
4. L-68, "supressed" should be "suppressed."
5. L-94, "enhance" should be "enhanced."
6. L-185, authors introduce "suntag:SDG2:SNC1-g4." What is the meaning of "g4"? Was it previously tested single guide RNA and found to be the best?

7. Figure 1e and S4 shows ST:SDG2:FVA_g4 line #10 to be more associated with H3K4me3 marks as compared to ST:SDG2:FVA_g4 line #12. It would have been informative to provide the RT-qPCR FVA expression for line 10 and 12 individually, and a sentence to whether more H3K4me3 deposition means higher expression, or any other comment on that observation.

8. Figure 2b, the x-axis legends say "#7" and "#1" which presumably represents individual lines, this would not be correct if the scatter plots on the bars represents individual lines. Authors should address this.

9. For figure-2a, is rdr6, bal, SunTag:SDG2:SNC1_g4 #7 and SunTag:dSDG2:SNC1_g4 #1 all taken as one snapshot, or they are multiple pictures patched up together. If they are multiple pictures patched up authors should provide magnification for each individual picture for clear demonstration of fitness effects.

10. L-864, "P4 seeds" but in other instances it is referred as Plant_4. Authors should ensure consistency and definition of labels.

11. The author did not provide the correct GEO accessions for raw data or the accessible GitHub page for the custom codes referenced on lines 571 and 573, respectively. Providing raw data and analysis methods is essential for reviewers to fairly assess the presented data.

Version 1:

Reviewer comments:

Reviewer #2

(Remarks to the Author)

I appreciate the authors' responses to my original comments. The new H3K4me3 and data analyses firmly support their major conclusions. This will make a nice contribution to the field.

Reviewer #3

(Remarks to the Author)

In this revision, the authors carried out additional experiments and analyses, which have clarified my concerns and questions. Although the systems require further improvement to achieve robust gene activation for more meaningful applications in plants, this study, with its comprehensive data and analyses, represents a novel and significant contribution to the research community and thus deserves publication in Nature Communications.

Point-by-point Response

We would like to thank the reviewers for their time and expertise in helping to review our manuscript. We feel that the manuscript has been substantially improved in addressing the thoughtful comments raised.

Below we provide a point-by-point response. Our responses are in **blue**.

Please note that in order for the line numbers referred to here to correspond to the correct place in the manuscript document, please ensure that the manuscript Word document is in the “No Markup” Tracking Mode.

Reviewer #1 (Remarks to the Author):

I read the manuscript of Binenbaum et al. with great interest. In this is a very carefully done study. The authors were able to show that SunTag-mediated targeting of H3K4me3 can influence different biological processes in Arabidopsis. They could show an enhancement of gene expression, which in their readout led to an increase in pathogen resistance. Even more exciting, they were able to change meiotic recombination patterns by the use of a chromosome 3 centromere repeat specific sgRNA. Although the effect was minor and the specificity due to the presence of many similar repeats limited, the fact that the authors were able to redirect recombination by a CRISPR/Cas based approach is notable. The results are novel, represent a valid proof of concept and of interest to a wide audience.

We are extremely grateful for this positive feedback. Thank You!

Reviewer #2 (Remarks to the Author):

Binenbaum and Adamkova et al present epigenome targeting of H3K4me3 using the plant SDG2 and the mammalian PRDM9 histone methyltransferases. They convincingly show that the SunTag system rather specifically targets these histone methyltransferases to their targets to activate gene expression. This was shown by targeting FWA and also SNC1 in Arabidopsis. It's fascinating that the dSDG2 can also reactivate FWA at a low frequency and it seems to me that PRDM9 might be able to as well (see comments below). I think this is a part of the story that is undersold and quite intriguing. This study also revealed increased frequency of recombination by targeting H3K4me3 to the centromeres, which is an important result given these regions of the genomes are somewhat recalcitrant to recombination in many plant species.

Overall, this is a really nice study that will pave the path forward to epigenome editing and improved recombination in plants. The comments below are intended to improve this already strong manuscript.

Thank you for the positivity! We feel that the updates and suggestions have further strengthened the manuscript and we appreciate the insightful feedback.

1. SunTag efficiently targets H3K4me3 to FWA in a gene specific manner. This result would be strengthened by adding ChIP-seq of H3K4me3 instead of PCR.

We agree and have now performed H3K4me3 ChIP-seq in the SunTag:SDG2 FWA-targeting lines. The data show clear enrichment at the FWA locus, extending into the gene body, closely mirroring the pattern observed with SunTag:PRDM9 (see new Fig. 1E as compared to 4E). To assess broader effects, we generated metaplot profiles over genes, TSSs, and endogenous H3K4me3 peaks (Fig. 1G). These reveal a modest reduction in signal at endogenous sites and a corresponding increase in background signal, consistent with some modest, non-specific off-target activity by the effector. Notably, the same pattern appears in the FWA targeting PRDM9 lines (just as the reviewer predicted) and the centromere targeting SDG2

lines (see Fig. 4F and Supp. Fig. 13A), suggesting this is a general feature of the SunTag coupled H3K4me3 methyltransferases, rather than an issue with ChIP quality. For additional QC, we now also include ‘fraction of read in peak’ (FRiP) scores and ‘fingerprint’ plots (Supp. Fig. 5, 13 and 20). In summary, H3K4me3 ChIP-seq data has been added, it confirms specific enrichment at FWA, akin to PRDM9, and allows us to quantify genome-wide off-target effects (see Fig 1 and lines 156).

2. Figure 3d. Enrichment of CEN targeting SDG2 looks great, but the H3K4me3 ChIP is not as convincing.

a. What is the quality of the ChIP-seq data? There is a lot more H3K3me3 in the euchromatin and centromere making this hard to interpret. Is it possible the ChIP did not work or is there off targeting of H3K4me3?.

We agree that in the previous version of the figure it was difficult to assess the quality and specificity of centromere targeted H3K4me3, as the euchromatic background enrichment in the chromosome arms obscured centromeric signal. To address this concern, we have revised the figure and performed several quality control and re-analysis steps:

1. ChIP-seq quality: We validated the quality of the H3K4me3 libraries by (i) including genome browser views with the H3K4me3 tracks from the SunTag and controls lines (Supp. Fig. 13B), (ii) showing metaplots of H3K4me3 over genes, TSSs, and endogenous peaks (Supp. Fig. 13A), and (iii) reporting FRiP scores and mapping statistics to provide quality control metrics (see Supp. Fig. 13CD).

2. Improved centromere analysis: We re-analysed H3K4me3 enrichment in the centromeric regions by focusing on specific peak regions identified in SunTag:SDG2:LRcen3 lines. In these lines, browser inspection revealed either novel or elevated peaks at centromeres compared to non-transgenic controls, while background signal in non-peak regions remained similar (see new Supp. Fig. 12). We then quantified H3K4me3 enrichment specifically over these peak regions, replacing the previous window-based averaging approach (Fig. 3D, Supp. Fig. 10C).

This updated analysis shows that SunTag:SDG2:LRCen3 lines exhibit reproducible, targeted enrichment of H3K4me3 at centromeres (see updated Fig 3D), consistent with the observed recombination phenotypes and the guide RNA targeting *in silico* analysis (Fig. 3AD and Supp. Fig. 11).

b. The replicate data in the supplement show a convincing results. I would add some standard metaplots of H3K4me3 around the TSS to show the quality of the ChIP data.

Thank you, and as suggested (and also mentioned above) we have now added metaplots of H3K4me3 over the TSS, gene bodies, and endogenous peaks (see Supp. Fig. 13A).

c. How do you reconcile the specificity in Figure 1 with the widespread off-targeting in Figure 3?

We believe that the apparent discrepancy has now been resolved through the addition of new data and revised analysis:

1. Comparable off-target H3K4me3 profiles: The newly generated H3K4me3 ChIP-seq data in Figure 1 (and as described above) reveal that the FWA-targeted SunTag:SDG2 lines also display modest off-target H3K4me3 enrichment, similar to that observed in the centromere-targeted lines (compare Fig. 1EG and Supp. Fig. 5&13). This suggests that the background activity is an inherent feature of the system rather than specific to any one target locus.
2. Refined centromere analysis: As described above, in Figure 3D, we have revised the analysis to focus specifically on peak regions within the centromeres that exhibit clear H3K4me3 enrichment in SunTag:SDG2:LRCen3 lines. The enrichment at these regions now clearly correlates with the SunTag-dCas9 binding profile, supporting the conclusion that the observed H3K4me3 signal reflects targeted activity rather than widespread off-targeting.

These updates reconcile the contrast between the previous version of Figures 1 and 3, and the extent of off-targeting is now discussed in detail in the new Discussion (line 478).

d. Where is the ectopic H3K4me3 located? Is it in gene bodies or TSSs. A closer inspection would be interesting to know for the readership.

We agree this is important for readership. To formally assess this we generated metaplots over gene bodies and TSSs as suggested (see above and Supp. Fig. 13A). The ectopic H3K4me3 generally accumulates more in the gene bodies and at non-specific off-target regions. This pattern of off-target is also virtually identical to what we see for FWA targeted SDG2 and PRDM9 lines. We discuss the findings of off-target effects in the Discussion (line 478).

3. The PRDM9 targeting of H3K4me3 is quite widespread throughout the genome body raising the possibility that it is either off-targeting or H3K4me3 is weakly enriched. Is it possible that PRDM9 can also initiate activation by recruiting additional factors similarly to SDG2?

We thank the reviewer for this insightful question. Our new ChIP-seq data (Fig. 1E) reveal that SunTag-SDG2 produces a gene-body H3K4me3 profile that closely mirrors the pattern generated by SunTag-PRDM9 (Fig. 4E). At FWA, maintenance of repressive DNA methylation continuously opposes the euchromatinising effect of H3K4me3, so the apparently “weak” enrichment most likely reflects this ongoing antagonism rather than inefficient targeting. The importance of tether positioning is underscored by the related study of Wang et al. 2025 (<https://doi.org/10.1038/s41477-025-01924-y>): two FWA targeting guides produced heritable late-flowering phenotypes with SunTag:SDG2, whereas our single-guide constructs for both SDG2 and PRDM9 did not, indicating that placement, not enzyme identity, often limits activation. Mechanistically, activation by both factors depends strongly on their methyltransferase activity: catalytic mutants abolish PRDM9-mediated FWA activation (Supp. Fig. 17) and sharply reduce - but do not completely eliminate - SDG2-mediated activation (Supp. Fig. 15, Supp. Fig. 16), leaving open the possibility of either co-recruitment or residual catalytic activity.

Taken together, the similar gene-body H3K4me3 profiles, strong dependence on catalytic activity, counteracting DNA-methylation pathways and guide-placement sensitivity all indicate that both PRDM9 and SDG2 activate FWA primarily through downstream readers of H3K4me3 that recruit DNA demethylases and/or enhance RNA polymerase II engagement. We have expanded on this in the Discussion to reflect this interpretation (from line 431).

4. Data should be deposited to GEO.

The data has now been deposited to GEO under the accession (GSE288686) and is available with the reviewer access token (ybobyqozfabtmt).

5. Please add better labels to the samples in Figure S7 as it's unclear what they currently refer to.

Thank you for pointing this out. We have updated these plots (now Supp. Fig. 10) to provide consistent meaningful labelling.

Reviewer #3 (Remarks to the Author):

In this study, Binenbaum et al. investigate the causal role of H3K4me3 in regulating transcription and meiotic recombination. By employing a CRISPR-based SunTag system guided by single guide RNAs, they introduce H3K4me3 marks at specific genomic loci using two histone modifiers, SDG2 and the mammalian PRDM9. Their results show that directing H3K4me3 deposition with SDG2 to the previously silenced *FWA* locus successfully activates transcription. They also demonstrate that targeting the *SNC1* disease resistance gene with SDG2 enhances its expression, thereby increasing *Arabidopsis* resistance to *Pseudomonas syringae*. Beyond gene activation, the authors show that applying H3K4me3 marks to centromeric regions, typically associated with very low recombination rates improves crossover frequency. Finally, by using PRDM9 as the H3K4me3 writer, off-target effects are reduced and the overall efficacy of the SunTag system is increased. While these findings present new tools for achieving precise epigenetic modifications in plants, important questions remain to be addressed.

We thank the reviewer for their appreciation and thoughtful feedback on the work.

Major Points:

1. It is important to show a strong causal relationship between increased H3K4me3 marks and the increased gene expression & phenotype. However, the current conclusion is compromised by 1) wide-spread off-target H3K4me3 deposition on a genome-wide scale, and 2) the use of only one or two selected lines to represent each construct (the gold standard is to use three independent transgenic lines). So, it is important to share data for all recovered lines to better understand the variability and efficiency of the system, and hence carefully draw conclusions from there. Since, In the methods section, the authors discuss that they have performed RT-qPCR on recovered 14-day-old whole seedlings with SDG2 experiments and all T1 lines with PRDM9. I recommend including a supplementary table or figure summarizing data from those transgenic lines for expression of the genes *FWA* and *SNC1*.

This is a very good idea. To address this, we carried out a screen of *FWA* targeting T₁ SunTag:SDG2 plants (Supp. Fig. 16). The results are similar to what we observe with PRDM9, with a high proportion of SunTag:SDG2 T₁s causing *FWA* activation (83%). As predicted from our previous results shown in (now Supp. Fig. 15) we observe that 14% of the no guide and 6% of the catalytic mutant SDG2 lines are able to activate *FWA* expression (Supp. Fig. 16). In contrast, 0% of the no guide and 0% of the catalytically dead PRDM9 T₁s activated *FWA* (Supp. Fig. 17). This provides support for the hypothesis that PRDM9 has reduced potential for off-target as compared to SDG2. Performing a similar analysis for *SNC1* in the T₁ generation is not feasible, as *SNC1* expression is known to be constrained (Stokes et al., 2002; Yang et al., 2023). Unlike *FWA*, which transitions from silent to highly expressed and is therefore well suited to T₁ screening, *SNC1* has a much narrower dynamic range. However, we now include data from a second independent line targeting *SNC1*, which shows increased *SNC1* expression, reduced rosette surface area, H3K4me3 enrichment, and enhanced resistance to *Pst::LUX* (see Supp. Fig. 7).

2. Since the authors claimed that PRDM9 is better than SDG2, it is necessary to have more comparisons of these two systems, which is needed to support the

author's claim. Spending the majority of the work only on SDG2, not the comparison of these two, does not help make this conclusion.

In this revised version of the manuscript, we provide several new comparisons of the two systems. As mentioned above, we carried out the T₁ screening of FWA targeting SunTag:SDG2, which provides strong support for the claim that PRDM9 has reduced off target potential (Supp. Fig. 16 versus Supp. Fig 17). We now also add histone westerns with SunTag-PRDM9 targeting and no guide lines, finding that - unlike SDG2 – PRDM9 lines that lack guide RNAs do not display any significant increase in bulk levels of H3K4me3 (Supp. Fig. 18). Again, this provides strong evidence in favour of PRDM9, and is consistent with the lack of pleiotropic defects observed in these lines. However, it is also important to note that many aspects of SDG2 and PRDM9 function are similar, including the patterning of H3K4me3 deposition at FWA, and the mild reduction of H3K4me3 at endogenous peaks at the metagene level, which is consistent with modest non-specific off-targeting (see Fig. 1 vs Fig. 4). Finally, we now also show that PRDM9 can similarly increase meiotic recombination over the centromeric region (see Fig 4G). These new data are included in the manuscript and we now discuss the comparisons between SDG2 and PRDM9 throughout the updated Discussion.

3. Is this epigenome stable across generations? Does the epigenome editing depend on the presence of transgene? These questions need to be answered in this manuscript.

The effect at FWA is not stable and does depend on the presence of the transgene. We assessed this by looking at T₂ plants segregating for the transgene finding no evidence for FWA expression in the absence of the transgene (Supp. Fig. 6). This is consistent with a lack of any plants switching to late flowering in the T₁, T₂ or T₃ generation plants for both PRDM9 and SDG2. It is worth noting that a related study using two guides to target FWA does cause a switch to late flowering (Wang et al 2025), suggesting that the single guide may result in insufficient, non-functional, mis-localised or mis-timed FWA transcripts unable to reach the critical threshold required to fully displace endogenous silencing machinery and switch FWA into the stably active epiallelic state. These new results are shown (Supp. Fig. 6) and are discussed in the Results (line 166) and new Discussion (line 447).

4. While the authors have demonstrated multiple use case scenarios of H3K4me3 depositions in Arabidopsis. Evidence shows low activation levels as demonstrated by qPCR and sequencing results. Since Arabidopsis has a small genome, this would raise a question to the applicability in crops, given their more complex genome and difficulty in transformations. Authors should at least discuss considerations and potential challenges while applying these tools/methods in higher plants other than Arabidopsis.

We agree that the applicability in crops is an important consideration. Notably SunTag has been successfully deployed for locus specific modulation of DNA methylation in rice (10.1016/j.cell.2025.04.036). We now include following sentence "However, off-target effects and metabolic impacts are important considerations when designing these systems for use in crops, which possess larger genomes and more complex regulatory architectures than *A. thaliana*" (see line 497).

5. For centromeric targeting of SDG2. Col-0 plants were crossed with CTL3.9 to get CTL3.9 F1 double hemizygous. This F1 was transformed with SunTag:SDG2 constructs. I am confused to why the authors are now leaving behind *rdr6* mutants. Initially the authors have benchmarked *rdr6* as the best for the expression of SunTag and used it in multiple sections of the paper until centromeric targeting. Why?

The reason for this is because it would not be practically feasible to obtain plants that are double hemizygous for *CTL3.9*, and homozygous for *rdr6*. The *rdr6* mutant is useful to reduce the impact of transgene silencing (Wang et al 2023) - so while it is the favoured where possible, it is not essential to carry out SunTag experiments. In line 565 we now explain this point clearly for readership.

6. In previous work, flowering time changes were observed because of FWA perturbations (DOI:10.1073/pnas.1716945115). Did H3K4me3 deposition, mediated by SDG2 and/or PDRM9, result in any observable phenotypes? Could you provide more details on these observations?

As mentioned above, we did not see any plants switch from early to late flowering in T₁, T₂ or T₃ generations. On interpretation is that the single guide may result in insufficient, non-functional, mis-localised or mis-timed *FWA* transcripts unable to reach the critical threshold required to fully displace endogenous silencing machinery and switch *FWA* into the stably active epiallelic state. Our companion study (Wang et al 2025) observed the same thing, whereby the switch to late flowering was only observed when a second *FWA* targeting guide was added. We now discuss this important point in our Discussion (line 447).

Minor Points:

1. The use of '#' annotations (e.g., #12) throughout the manuscript is unclear and inconsistent. For example, in Figure 1c, 'SunTag:SDG2:FWA_g4 #12' could be interpreted as a line number, but the legend says error bars represents three biological replicates, which would mean #12 is not a line number. Clarify the meaning of '#' annotations across all figures and text.

In Figure 1C the biological replicates are pools of 10 seedlings from the individual line. The legend now reads "Error bars represent SEM from three biological replicates (a pool of 10 seedlings) from the lines (#) and genotypes indicated". We have updated the figure legends throughout to remove this ambiguity.

2. For Figure 3c, the control should have been CTL3.9 F1 double hemizygous lines crossings with no activation instead of Col-0 crossed with CTL3.9. Even though this control is provided in the supplementary material, it should also be included in Figure 3c for clarity and better representation. This is because CTL3.9 F1 double hemizygous lines form the background used to improve recombination frequency and already have a partially disturbed centromeric region of chromosome 3 due to the GFP-dsRed insertions. Recombination with Col-0 plants would be biased, as their centromere and pericentromeric regions are not disturbed, and they naturally exhibit low recombination rates.

We acknowledge that the no guide SunTag:SDG2 is a relevant control and have moved it to Figure 3C. We also take the point that there will be some disturbances from the T-DNA insertions. That said, in Fernandes et al., 2024, we investigated this CTL3.9 line in detail, finding no evidence for significant rearrangements as compared to Col-0. The insertion sites were also mapped, with the green (CG17) and red (CR55) insertions spanning 12.7-Kb and 5.1-Kb, respectively, while the interval itself spans nearly 9-Mb. Therefore, the T-DNA insertions are not likely to have an effect when measuring recombination events at this scale. Furthermore, as this is an inbred experiment, the Col-0 control also has the double hemizygous T-DNA's, and so provides the baseline to compare against. We have updated the manuscript to make this point more clearly (line 562) and have also updated Figure 3C to include the additional control in the main figure as suggested.

3. L-40, grammar correction “its”(not “their”) performance.

Corrected, thank you.

4. L-68, “supressed” should be “suppressed.”

Corrected, thank you.

5. L-94, “enhance” should be “enhanced.”

Corrected, thank you.

6. L-185, authors introduce “suntag:SDG2:SNC1-g4.” What is the meaning of “g4”? Was it previously tested single guide RNA and found to be the best?

Thank you - this is simply a carryover from our internal guide naming system. Guides were not systematically compared - this was guide was predicted to bind closest to the TSS. We have added this information to the manuscript (line 540).

7. Figure 1e and S4 shows ST:SDG2:FWA_g4 line #10 to be more associated with H3K4me3 marks as compared to ST:SDG2:FWA_g4 line #12. It would have been informative to provide the RT-qPCR FWA expression for line 10 and 12 individually, and a sentence to whether more H3K4me3 deposition means higher expression, or any other comment on that observation.

As suggested by the reviewer, we performed FWA RT-qPCR on lines 10 and 12 grown side-by-side, and find no significant difference in their levels of FWA activation (see Supp. Fig. 4B and figure legend).

8. Figure 2b, the x-axis legends say “#7” and “#1” which presumably represents individual lines, this would not be correct if the scatter plots on the bars represents individual lines. Authors should address this.

Yes, the # signs do represent individual lines, and the dots represent measurements from the individual plants of those lines. We have updated the figure legend (“Each dot represents an individual plant” - see Fig 2B, line 239) to make this more clear.

9. For figure-2a, is rdr6, bal, SunTag:SDG2:SNC1_g4 #7 and SunTag:dSDG2:SNC1_g4 #1 all taken as one snapshot, or they are multiple pictures patched up together. If they are multiple pictures patched up authors should provide magnification for each individual picture for clear demonstration of fitness effects.

To generate the data for leaf surface area plot shown in Fig 2B, pots containing 4 to 5 plants were imaged individually for measurement. Shown in Figure 2A are cropped representative plants from those images, which we have now separated and added appropriate scale bars (see updated Fig 2A).

10. L-864, "P4 seeds" but in other instances it is referred as Plant_4. Authors should ensure consistency and definition of labels.

Great spot, thank you. We have updated to P4 line throughout to ensure consistency.

11. The author did not provide the correct GEO accessions for raw data or the accessible GitHub page for the custom codes referenced on lines 571 and 573, respectively. Providing raw data and analysis methods is essential for reviewers to fairly assess the presented data.

The NGS data has been deposited to GEO under the accession (GSE288686) and is available with the reviewer access token (ybobyqozfabmt). We have also now made the GitHub page publicly available (<https://github.com/C-Jake-Harris/Binenbaum-SunTag-H3K4me3>)